# Cantor🪄: Inspiring Multimodal Chain-of-Thought of MLLM

## ABSTRACT

With the advent of large language models (LLMs) enhanced by the chain-of-thought (CoT) methodology, visual reasoning problem is usually decomposed into manageable sub-tasks and tackled sequentially with various external tools. However, such a paradigm faces the challenge of the potential "determining hallucinations" in decision-making due to insufficient visual information and the limitation of low-level perception tools that fail to provide abstract summaries necessary for comprehensive reasoning. We argue that converging visual context acquisition and logical reasoning is pivotal for tackling visual reasoning tasks. This paper delves into the realm of multimodal CoT to solve intricate visual reasoning tasks with multimodal large language models (MLLMs) and their cognitive capability. To this end, we propose an innovative multimodal CoT framework, termed Cantor, characterized by a perception-decision architecture. Cantor first acts as a decision generator and integrates visual inputs to analyze the image and problem, ensuring a closer alignment with the actual context. Furthermore, Cantor leverages the advanced cognitive functions of MLLMs to perform as multifaceted experts for deriving higher-level information, enhancing the CoT generation process. Our extensive experiments demonstrate the efficacy of the proposed framework, showing significant improvements in multimodal CoT performance across two complex visual reasoning datasets, without necessitating fine-tuning or ground-truth rationales.

## CCS CONCEPTS

• **Information systems → Question answering**.

## KEYWORDS

Multimodal Chain-of-Thought, Visual Reasoning

## 1 INTRODUCION

With the development of large language models (LLMs), researchers have begun to adopt the chain-of-thought (CoT) strategy to improve the model performance in reason tasks. CoT mimics the gradual reasoning process of humans, helping models improve their deep understanding and analytical abilities by constructing a series of logical steps to solve complex visual reasoning problems. The effectiveness of CoT has been widely validated in language reasoning tasks. Recently, researchers have naturally extended its application to multimodal domains. Visual reasoning tasks [29, 30] are

*ACM MM, 2024, Melbourne, Australia*
© 2024 Copyright held by the owner/author(s). Publication rights licensed to ACM.
ACM ISBN 978-x-xxxxx-xxxx-x/YY/MM
https://doi.org/10.1145/nnnnnnn.nnnnnnn

**Unpublished working draft. Not for distribution.**

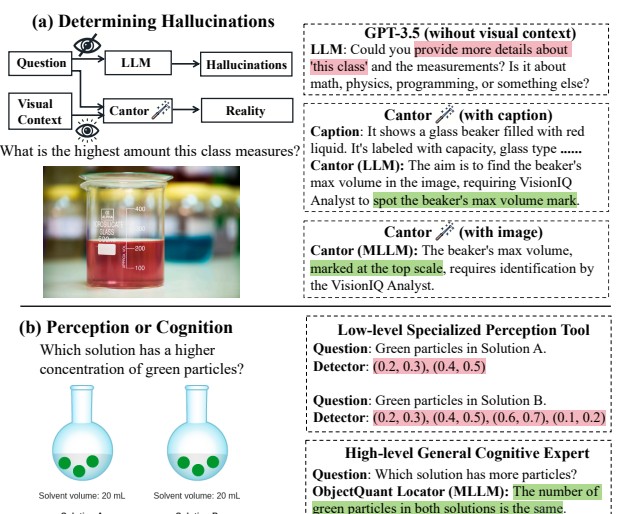

**Figure 1: (a) Comparison of visual information on decision generation:** Asking GPT-3.5 (without visual context) leads to "determining hallucinations" due to lacking clarity of the image. Cantor (with caption) by introducing visual context through captions, does not encounter this issue. Cantor (with image) is even more precise, improving the rationality of task assignment. **(b) Comparison of different visual tools:** Low-level specialized perception tools used in traditional approaches only obtain basic data. High-level general cognitive expert acted by MLLM obtains object number relationships, enabling direct and subsequent reasoning.

inherently suited for chain-of-thought (CoT) methodologies. These tasks necessitate that models not only "perceive" the contents and contexts within images but also "comprehend" these visual elements to make coherent inferences and decisions. Consequently, the exploration of multimodal CoT has significantly expanded in the research community.

Most existing multimodal CoT methods are divided into two stages: decision-generation and execution. 1)**Decision-Generation**. It is the first step in multimodal CoT methods, which involves understanding, analyzing, and formulating inference plans for the problem. The existing determining methods include breaking down problems into sub-problems [53], capturing scene maps in images [32], finding similarities and differences in related images [49], and so on [41, 44]. They attempt to simplify the problem at the textual level or add more contextual information at the visual level. 2) **Execution**. In this stage, models perform specific operations scheduled by the previous determining stage. Specifically, the model transforms the planning into practical solutions. The existing execution methods usually rely on various specialized API tools or vision-language models (VLMs), with the former emphasizing the specificity of task

execution [31, 41] and the latter emphasizing the universality of task execution [44, 53].

Although these multimodal CoT methods have improved the performance in visual reasoning tasks, there are still limitations: Firstly, when making decisions, existing methods often directly input plain text into LLMs without considering visual context [17, 44, 53]. Intuitively, this increases the divergent thinking of LLMs towards problems, but in reality, it may lead to "determining hallucinations". As shown in Fig. 1 (a), if the question itself is not closely related to the image and only asks "What is the highest amount this class measures?" based on the text, LLM (GPT-3.5) is not clear about what "this class" specifically means. It will answer that the provided information is insufficient and begin to guess whether the "class" refers to a metric in physics or a class in programming. This perception uncertainty may lead LLMs to make decisions that are unrelated to the problem or even incorrect, misleading subsequent execution and resulting in completely unrelated answers.

Secondly, during execution, existing methods typically execute tasks by calling external tools, because MLLMs still fall short of solving numerous visual reasoning tasks [17, 31, 32, 38, 44]. But these tools are mostly low-level visual-perception tools (detectors, recognizer, OCR, etc.) that can only extract low-level visual information. As shown in Fig. 1 (b), when comparing the number of particles in solutions, they only provide the positions of particles and fail to infer high-level information such as the relationship between their numbers. They further input these low-level clues into LLMs for organization and summarization [17, 32, 53]. When complex clues increase, this undoubtedly increases the burden of LLMs on long-text reasoning. Meanwhile, with many external tools, it also increases the complexity of the pipeline.

To address the above limitations, we propose a novel multimodal CoT framework, Cantor. In decision generation, we enable an MLLM or an LLM to act as a cantor within the chorus, simultaneously processing visual and textual context for comprehensive understanding, and then assigning specific tasks to "experts" acted by a single MLLM for high-level logical problem-solving. Specifically, during the decision generation, we analyze in detail the importance of visual information in the determining stage. This includes the quality of determining with or without visual information, as well as the differences in the impact of detailed or concise visual information on determining. Ultimately, we conclude that visual information is crucial during the decision generation stage. When we use an MLLM model (such as Gemini) for the decision generator, we directly feed images into the model to fully comprehend the question and deliberate on it. However, when employing an LLM model (such as GPT-3.5), we find that providing a more detailed caption of the image is more conducive to understanding the question. Furthermore, the decision generator is required to explicitly provide explanatory decisions, including problem-solving strategies, reasons for expert invocation, and specific task conduction for each expert. Consequently, it guides an MLLM to act as tailored experts (such as ObjectQuant Locator, TextIntel Extractor, VisionIQ Analyst, and ChartSense Expert) to provide conclusive answers for sub-tasks in the process. As shown in Fig. 1 (a), when using LLM to make a decision, with detailed caption guidance, the model knows that it is asking for the maximum volume of the beaker and makes the correct decision. The decision is clearer when the image

is available to the MLLM, that is, requiring the VisionIQ Analyst to extract the number at the top of the cup wall.

During execution, we observe that MLLM is an advanced cognitive tool that performs better in directly acquiring high-level information (e.g., relative position and quantity) than acquiring low-level visual information like detecting positions. Such high-level information is superior for multimodal CoT. Instead of using several external tools, Cantor assigns different tasks to a single MLLM via different expert identities and task instructions, exploring the professional potential of an MLLM acting as certain experts. The tailored experts provide high-level professional information directly, thus reducing the burden of subsequent integrated reasoning. As shown in Fig. 1 (b), when comparing the concentration of green particles, we need to compare the number of particles in the two bottles first. MLLM acts as an ObjectQuant Locator and directly compares the quantity variance in the two solutions. Compared with obtaining the position of particles, MLLM gets the result of the quantity relationship more accurately. This result is directly applied to the further inference of the final answer.

Our proposed framework Cantor achieves SOTA results in both ScinceQA [30] and Mathvista [29]. When Gemini is used as the decision generator, Cantor obtains an accuracy gain of 4.11% and 5.9%, respectively. Employing GPT-3.5 in Cantor also achieves an accuracy gain of 2.24% and 9.2%. In all of our experiments, we use only one MLLM (Gemini) to play the role of multiple experts, performing different sub-tasks with different requirements. Our contributions are the following:

- We propose an inspiring multimodal CoT framework named Cantor, which features a perceptual decision architecture that effectively integrates visual context and logical reasoning to solve visual reasoning tasks.
- We utilize the advanced cognitive abilities of an MLLM to act as multifaceted experts, obtaining higher-level information and significantly enhancing CoT generation.
- We demonstrate Cantor's effectiveness on two challenging benchmarks, largely surpassing existing counterparts.

## 2 RELATED WORK

### 2.1 Multimodal Large Language Models

Recent researches indicate that the development of Multimodal Large Language Models (MLLMs) [6, 10, 11, 33, 37, 39, 47, 48] is the result of combining the advanced reasoning capabilities of Large Language Models (LLMs) with the capabilities of Vision-Language models (VLMs). These models have achieved significant performance improvements in multimodal tasks by integrating visual and linguistic information. In particular, significant progress [13, 24, 36]has been made in connecting visual and text representations with contrastive visual and language models, but they encounter limitations when dealing with downstream tasks that require generating components or performing more refined reasoning on visual and language. To overcome these limitations, MLLM extends the reasoning and generation capabilities of LLM to the visual domain by directly inferring embedded visual features [1, 2, 7, 9, 23, 54]. In addition, MLLMs further improve performance through fine-tuning visual instructions [28].

These advances not only demonstrate the ability of MLLM to handle complex multimodal information but also provide new possibilities for achieving General Artificial Intelligence (AGI) with rich multimodal information. By integrating the text reasoning ability of LLM with the image understanding ability of visual language models, MLLM can achieve deep understanding and expression in multiple modalities, processing complex tasks such as image captioning and visual question answering. Open-source MLLMs such as LLaVA [28] demonstrate these capabilities, while closed-source models such as GPT4-V [34] and Gemini [40] have taken a greater step in capturing scene context, reasoning, and creativity. Although for specific tasks these closed-source models may not be directly competent or fine-tuning. However, prompt learning can to some extent overcome these limitations. This paper is dedicated to exploring the technique of CoT [43] to enhance the ability of MLLMs to capture the complete context of complex visual scenes, thereby further strengthening their reasoning capabilities.

## 2.2 Tool-Augmented Language Models

In recent years, despite the impressive performance of Large Language Models (LLMs), they are not without their inherent limitations. These include challenges such as obtaining up-to-date information [21], the inability to employ specific tools [31, 38], and difficulties in executing complex reasoning processes [29, 30]. Meanwhile, researchers are increasingly interested in using external tools and modular methods to enhance LLM through prompting and in-context learning. These enhanced LLMs can utilize different external tools to provide LLMs with more functionality and gain more knowledge. Some works [5, 12, 17, 19] utilized prompts to generate complex programs that can be executed by computers, calling different tools to more effectively perform logical reasoning tasks. For example, PaLI-X-VPD [17] extracted the reasoning ability of LLM by generating multiple candidate programs, executing programs through external tools, and verifying their correctness. It transformed each correct program into a language description of reasoning steps to form a CoT. In addition, some works proposed benchmarks (such as API Bank [25], ToolQA [55], and Meta-Tool [18]) to evaluate the effectiveness of LLM tool use. This article mainly emphasizes enhancing the tool usage ability of MLLM.

## 2.3 Multi-modal CoT Reasoning

LLMs and MLLMs are becoming increasingly popular. Although their own abilities are becoming stronger, good prompt methods are still the key to fully unleashing their abilities. Chain-of-thought (CoT) is a method to improve LLM's reasoning ability, and the core of CoT is to encourage LLM to clarify their reasoning in a human thinking way, specifically by adding logical thinking processes before obtaining answers. In the field of NLP, CoT has received extensive research [8, 15, 42, 51]. Jason Wei *et al.* [43] significantly improved LLM's reasoning ability by simply adding problem-solving ideas directly to in-context examples. Subsequently, researchers mainly focused on how to automate the construction of CoT to reduce manual annotation and more complex structures such as Tree-of-Thought (ToT) [45] and Graph-of-Thought (GoT) [3, 22, 46].

Meanwhile, surprising progress has been made in multimodal CoT. MM-CoT [52] firstly proposed a two-stage reasoning framework by using text and image pairs as input, generating rationale first and then generating answers. Subsequent works [14, 14, 41, 53] are mostly based on this framework, focusing on designing special vision-language feature fusion mechanisms to enhance multimodal information interaction. However, these CoT prompting methods need to fine-tune on ground truth of natural language reasoning, which requires both annotation and computation costly. Based on this issue, researchers have proposed other CoT methods that do not require manual annotation and training. On the one hand, they fully tap into textual information. For example, DD-CoT [53] further refined the process of generating the CoT. Without introducing visual information, it used LLM to break down the problem into multiple related sub-questions and then answer each sub-question one by one to form the CoT. On the other hand, researchers are committed to enhancing visual information through various means. For example, CoCoT [49] captured image characteristics by comparing the similarities and differences between images, while CCoT [32] obtained scene maps by disassembling the targets and attributes in the images to assist in rationale generation. The key difference between our method and these methods is that when mining text information, we introduce visual information in advance to make decisions more reasonable and factual. In addition, we enhance visual information more comprehensively by calling multiple experts. Last, Cantor is also a method that does not require training or manual annotation, so it has strong universality and convenience. This paper emphasizes enhancing the expert usage capability of MLLM. Considering that MLLM has multimodal universal capabilities, it is naturally suitable to serve as various experts. Therefore, this paper will endow MLLM with various identities and explore its expert-playing abilities.

## 3 METHOD

To address the limitations of multimodal CoT in solving visual reasoning tasks, we propose Cantor, which introduces visual information to make correct decisions and uses a single MLLM to act as multiple experts to adapt to a wide range of problems. We describe the framework of Cantor (Section 3.1). Then, we provide a detailed introduction to our two-step approach: the first is Decision-Generation (Section 3.2), and the second is Execution (Section 3.3).

## 3.1 Preliminaries

Cantor consists of two stages: Decision-Generation and Execution, as shown in Fig. 2. During the Decision-Generation stage in Cantor, Cantor's input consists of $X = \{I, T, P_{in}\}$, where $I$ denotes the visual input (image or a caption), $T$ signifies the text input, which represents the concatenation of the problem statement and its context, and $P_{in}$ represents the prompt for generating decisions. Formally, given an input query $X$, a decision $P$ is generated as follows: $P_{out} = F(X)$, where $F$ denotes the decision generator (an LLM or MLLM). Specially, $P_{out} = \{R, O, S_t\}$, where $R$ denotes Principle Analysis, $O$ denotes Module Selection & Reason, and $S_t$ denotes the tasks assigned to expert modules. For specific examples, please refer to the blue section in the middle of Fig. 2.

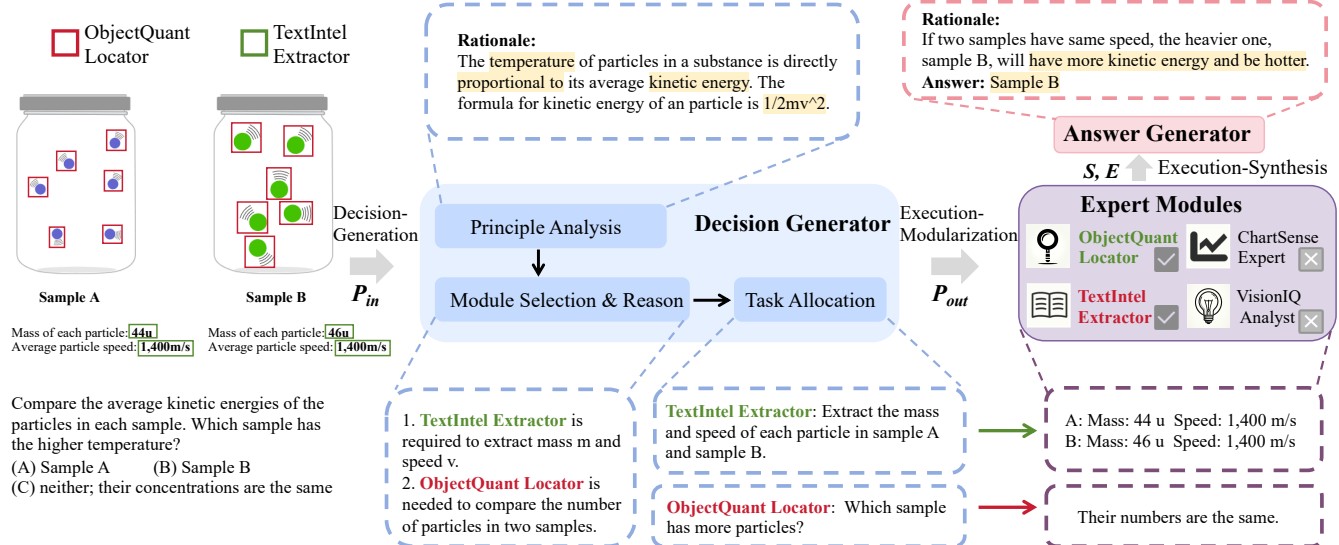

**Figure 2: Overview of Cantor and a specific example. Cantor analyzes the image and problem through the Decision Generator, offering the principle analysis of the questions, and providing module selection & Reason, as well as specific task allocation. Subsequently, MLLM acts as various expert modules to execute sub-tasks. Finally, Cantor synthesizes and contemplates through the Answer Generator, providing the final answer.**

In the execution-modularization stage, multiple sub-tasks $S_t = \{st_1, st_2...st_n\}$ derived from the decision $P_{out}$ and image $I$ are jointly sent to the corresponding expert module to obtain the sub-answers $S_a = \{sa_1, sa_2, ..., sa_n\}$. The process is as follows: $S_a = G(S_t, I)$, where $G$ denotes various experts (an MLLM). This process corresponds to the Execution-Modularization stage in the purple section at the bottom right of Fig. 2. Then in Execution-Synthesis stage, we concatenate the sub-tasks and sub-answers to form supplementary information $S = \{S_t, S_a\}$, and design an answer generation prompt $E$. Finally, feed the updated input $X' = \{I, T, S, E\}$ and infer the final answer $A = F(X')$, where $F$ denotes the answer generator (an LLM or MLLM), as shown in the upper right corner of Fig. 2.

## 3.2 Step 1: Decision-Generation

Our first step is to generate decision $P_{out}$ which considers and deploys the problem. Please note that we are studying unsupervised visual reasoning tasks, which involve having the model generate corresponding decisions for the problem without ground truth [44, 49]. Additionally, for standardization and accuracy, we adopt a few-shot setting in prompt to provide a decision generation prompt $P_{in}$ for the model, which includes the requirements for decision generation, the characteristics of callable modules, and several manually written decision examples.

Let's provide a detailed introduction to the Decision-Generation process of Cantor and the specific components of the prompt $P_{in}$:

**1. Acting as Decision Generator.** We prompt the LLM or MLLM with "You are an advanced question-answering agent required with four specialized modules to aid in the analysis and responding to queries about images" enabling it to function as a decision generator in Cantor.

**2. Expert Modules Unveiled.** As shown in the Expert Modules of Fig. 2. We provide detailed information on the characteristics of each expert module for Cantor, with the aim to allocate tasks to each expert module based on the principle of addressing the problem during the Decision-Generation phase, as follows: **TextIntel Extract:** This module extracts and converts text within images into editable text format. It's particularly useful for images containing a mix of text and graphic elements. **ObjectQuant Locator:** This module identifies and locates objects within an image. It's advanced at comparing quantities and recognizing spatial relationships. **VisionIQ Analyst:** This module processes and interprets visual data, enabling you to ask any queries related to the image's content. **ChartSense Expert:** This module specializes in analyzing and interpreting information from charts and graphs. It can extract data points, understand trends, and identify key components such as titles, axes, labels, and legends within a chart.

**3. Principle Analysis and Module Selection & Reason.** We prompt Cantor "Provide a rationale for your approach to answering the question, explaining how you will use the information from the image and the modules to form a comprehensive answer", performing an overall assessment and modular analysis of the question.

**4. Task Allocation.** We prompt "Assign specific tasks to each module as needed, based on their capabilities, to gather additional information essential for answering the question accurately.", requiring Cantor to select the necessary modules and assign their corresponding specific tasks.

**5. Contextual Insights and Practical Applications.** We introduce some in-context examples to enhance Cantor's comprehension of our prompts, ensuring its responses adhere to the desired format. Detailed instances are provided in the supplementary materials for further reference. Then, we input the particular problem that needs

addressing, along with its contextual details, enabling Cantor to formulate nuanced decisions. The blue part on the left half of Fig. 2 shows a specific example of decision generation.

The above five parts are combined to form the final decision generation prompt $P_{in}$. Subsequently, $P_{in}$ together with visual input $I$ and text input $T$, constitutes the complete input for the first stage of Cantor, prompting Cantor to deliver a deliberate decision $P_{out}$.

The decision generation method represents a core novel contribution of our work. Initially, the LLM or MLLM is employed as a decision generator, serving as the brain. Next, a suite of specialized expert modules is integrated, augmenting the decision generating with diverse capabilities analogous to the limbs. This integration ensures that decision-generating is both comprehensive and granular, leveraging the strengths of each module. Thereafter, the decision generator tailors tasks for selected expert modules based on insights gained from principle analyses. This dynamic task allocation enhances Cantor's efficiency and effectiveness. Ultimately, the introduction of in-context examples enables the MLLM to learn and reference, thereby further improving the accuracy and adaptability of decision generation. Notably, we introduce visual context in advance during the Decision-Generation stage, rather than the Execution stage, effectively alleviating determining hallucinations.

### 3.3 Step 2: Execution

In Cantor, the execution stage can be divided into two stages, Execute-Modularization and Execute-Synthesis. The former completes the sub-tasks assigned during the Decision-Generation stage by calling various expert modules and providing supplementary information. The latter summarizes various supplementary information from the execute-modularization stage and generates the final answer through rational and detailed thinking.

**Execute-Modularization.** We call the expert module to execute the various sub-tasks assigned during the Decision-Generation stage. Specially, we first extract sub-tasks $S_t = \{st_1, st_2...st_n\}$ from $P_{out}$. Next, we find the expert module corresponding to the sub-task $st_i$ in sequence, and input the sub-task $st_i$ as the prompt into the expert, such as "ObjectQuant Locator: Which sample has more particles?". Subsequently, we obtain the sub-task answer $sa_i$, such as "Their numbers are the same", as shown in the lower right part of Fig. 2.

Symbolically, we input the experts played by MLLM, sub-task $st_i$, and image $I$, and MLLM provides the execution results of the sub-task. The process is as follows: $sa_i = G(I, st_i)$, where $G(\cdot)$ represents MLLM acting as experts, and $sa_i$ represents the sub-task's answer. When executing sub-tasks, we only use one MLLM to act as different expert modules. This not only simplifies the pipeline of the method but also aims to fully utilize the advanced cognitive abilities of MLLM.

**Execute-Synthesis.** We concatenate and summarize the obtained sub-tasks and sub-tasks answers to obtain supplementary material $S$ for auxiliary reasoning, as follows: $S = \{[st_1, sa_1] \cdot [st_2, sa_2] \cdot ... \cdot [st_n, sa_n]\}$. Notably, in the answer generation stage, we introduce the answer generation prompt $E$, which includes the prompt and the formatting requirement for generating answers, as follows: "You are a knowledgeable and skilled information integration science expert. Please gradually think and answer the questions based on the given questions, options, and supplementary information. Please note that we not only need answers but more importantly, we need rationales for obtaining answers. Please combine your knowledge and supplementary information to obtain reasoning and answers. Please prioritize using your knowledge to answer questions. If unable to answer, maintain critical thinking and select effective information to assist you in selecting the most correct option as the answer. Furthermore, please do not rely solely on supplementary information, as the provided supplementary information may not always be effective."

This includes three key points. Firstly, we use prompts to have Cantor play the role of an answer generator who is knowledgeable and skilled at integrating information. This not only ensures its professionalism and ability to make basic judgments on questions but also ensures that it can better integrate information obtained during the Execute-Modularization stage. Secondly, to increase interpretability, demonstrate the thinking process of Cantor, and improve its thinking ability, we require Cantor to answer the basic principles first, and then generate the corresponding options, as shown in the pink box in Fig. 2. Finally, we request that Cantor remain rational and critical, ensuring it does not solely rely on the information obtained from the Execute-Modularization stage. This approach promotes a more balanced and comprehensive execute-synthesis process.

## 4 EXPERIMENTS

In this section, we evaluate the proposed Cantor on two visual reasoning datasets: ScienceQA [30] and MathVista [29]. The experimental results show that Cantor outperforms existing baselines in these tasks. Additionally, we analyze the importance of visual information in visual reasoning tasks. Finally, we conduct a detailed analysis of Cantor's key components.

### 4.1 Datasets

We evaluate our method on two visual reasoning task benchmarks.

**ScienceQA** [30]: It is the first multimodal scientific question-and-answer dataset annotated with detailed explanations. The problems with datasets are systematically divided into three main scientific disciplines: natural sciences (NAT), social sciences (SOC), and language sciences (LAN). We only use the ScienceQA test set, which contains 4241 questions and answers, of which 2,017 samples are attached with images.

**MathVista** [29]: It is a dataset that combines the challenges of various mathematical and visual tasks. It requires high levels of model granularity, deep visual understanding, and combinatorial reasoning ability, making it a challenging dataset for current basic models. In the experiment, we used Mathvista testmini, which includes 1000 text and image pairs for Q&A.

### 4.2 Models

We use two models to evaluate our method, GPT-3.5 and Gemini Pro 1.0, by calling their official API. Firstly, we use GPT-3.5 to evaluate the impact of introducing high-level perceptual information on LLM inference ability and explore the linkage ability between LLM and MLLM. Secondly, we use Gemini Pro 1.0, an advanced MLLM.

**Table 1: Accuracy scores (%) on ScienceQA [30], where bold entries indicate the best results, underlines indicate the second-best. We compare the performance of our system with various baseline models including supervised models and unsupervised models. Question classes: NAT = natural science, SOC = social science, LAN = language science, TXT = text context, IMG = image context, NO = no context, G1-6 = grades 1-6, G7-12 = grades 7-12.**

| Methods | Supervised | IMG | NAT | SOC | LAN | TXT | NO | G1-6 | G7-12 | Avg |
|---|---|---|---|---|---|---|---|---|---|---|
| Random Chance | ✗ | 40.08 | 40.28 | 46.13 | 29.25 | 47.45 | 33.66 | 39.35 | 40.67 | 39.83 |
| Human Average [30] | ✗ | 87.50 | 90.23 | 84.97 | 87.48 | 89.60 | 88.10 | 91.59 | 82.42 | 88.40 |
| UnifiedQA [20] | ✓ | 61.38 | 68.16 | 69.18 | 74.91 | 63.78 | 77.84 | 72.98 | 65.00 | 70.12 |
| UnifiedQA (CoT) [20] | ✓ | 66.53 | 71.00 | 76.04 | 78.91 | 66.42 | 81.81 | 77.06 | 68.82 | 74.11 |
| Multimodal-CoT [52] | ✓ | 82.90 | 87.52 | 77.17 | 85.82 | 87.88 | 86.83 | 84.65 | 85.37 | 84.91 |
| LLaMA-Adapter [50] | ✓ | 80.32 | 84.37 | 88.30 | 84.36 | 83.72 | 86.90 | 85.83 | 84.05 | 85.19 |
| LLaVa [28] | ✓ | 88.00 | 90.36 | 95.95 | 88.00 | 89.49 | 90.66 | 90.93 | 90.90 | 90.92 |
| LLaVA (GPT-4) [28] | ✓ | 88.99 | 91.56 | 96.74 | 91.09 | 90.62 | 93.52 | 92.73 | 92.16 | 92.53 |
| LLaMA-SciTune (CTOM) [16] | ✓ | 86.67 | 89.30 | 95.61 | 87.00 | 93.08 | 91.75 | 84.37 | 91.30 | 90.03 |
| GPT-3 (zero-shot) [4] | ✗ | 65.74 | 75.04 | 66.59 | 78.00 | 74.24 | 79.58 | 76.36 | 69.87 | 74.04 |
| GPT-3.5 (CoT) (AE) [35] | ✗ | 66.09 | 76.60 | 65.92 | 77.55 | 75.51 | 79.58 | 78.49 | 67.63 | 74.61 |
| GPT-3.5 (CoT) (ALE) [35] | ✗ | 67.43 | 75.44 | 70.87 | 78.09 | 74.68 | 79.93 | 78.23 | 69.68 | 75.17 |
| GPT-3.5 CoT [33] | ✗ | 67.92 | 78.82 | 70.98 | 83.18 | 77.37 | 86.13 | 80.72 | 74.03 | 78.31 |
| QVix(GPT-3.5) [44] | ✗ | 55.00 | - | - | - | - | - | - | - | - |
| Chameleon (GPT-3.5) [31] | ✗ | 70.80 | **81.62** | 70.64 | **84.00** | 79.77 | 86.62 | 81.86 | 76.53 | 79.93 |
| DD-CoT(GPT-3) [53] | ✗ | 69.96 | 78.60 | 73.90 | 80.45 | 77.27 | 82.93 | 80.65 | 73.50 | 78.09 |
| DD-CoT(GPT3.5) [53] | ✗ | 72.53 | 80.15 | 76.72 | 82.82 | 78.89 | 85.02 | 82.86 | 75.21 | 80.15 |
| Cantor(GPT-3.5) | ✗ | **77.54** | 80.37 | **85.49** | **84.00** | 77.27 | **86.83** | **85.61** | **76.60** | **82.39** |
| Gemini | ✗ | 76.85 | 79.13 | 85.26 | 80.82 | 76.93 | 83.83 | 83.81 | 75.54 | 80.85 |
| Cantor(Gemini) | ✗ | **82.40** | **84.24** | **87.85** | **84.09** | **82.11** | **86.97** | **88.18** | **79.17** | **84.96** |

We desire to fully tap into the multimodal ability of MLLM and improve its reasoning ability.

### 4.3 Implementation Details

We implement two versions of Cantor based on GPT-3.5 and Gemini. Cantor(GPT-3.5) uses both GPT-3.5 as the Decision Generator and Answer Generator during the Decision-Generation and Execute-Synthesis stage. Differently, Cantor(Gemini) uses Gemini in these two stages. For the Execute-Modularization stage, due to the need for multimodality, we use Gemini as the MLLM in both versions, playing various roles as experts. For the captions required for Cantor(GPT-3.5) in the Decision-Generation stage, we generated them through Gemini Pro 1.0, with the prompt "Please provide the detailed title of this image as much as possible". In terms of models' prompts, although the two models have different preferences for prompts, we use the same prompt for the sake of method universality in Decision-Genetation stage and Execute-Synthesis stage. The prompt in Execute-Modularization stage is generated by the Cantor itself. For different datasets' prompts, we design different in-context examples based on their characteristics, and the rest of the prompts are the same.

### 4.4 Main Results

**ScienceQA.** Tab. 1 shows the results of existing baselines compared to our method Cantor on ScienceQA. Using GPT-3.5 as the base LLM to decision and answer, Cantor achieves an accuracy of 82.39%, which is an improvement of 4.08% over the chain-of-thought (CoT) prompted GPT-3.5 [33]. Furthermore, with Gemini as the decision

**Table 2: Accuracy scores (%) on ScienceQA for the IMG class, which includes image context.**

| Method | Subject | | | Grade | | Average |
|---|---|---|---|---|---|---|
| | NAT | SOC | LAN | G1-6 | G7-12 | |
| LLaVA | 37.0 | 61.5 | 33.3 | 52.3 | 30.5 | 46.2 |
| MiniGPT | 45.2 | 51.5 | 38.1 | 50.6 | 39.1 | 47.4 |
| InstructBLIP | 43.9 | 58.1 | 47.6 | 53.1 | 39.4 | 49.3 |
| QVix (GPT-3.5) | 48.0 | 67.1 | 38.1 | 60.6 | 40.5 | 55.0 |
| Qwen-VL-Chat | - | - | - | - | - | 68.85 |
| mPLUG-Ow12 | - | - | - | - | - | 68.75 |
| Chameleon (GPT-3.5) | - | - | - | - | - | 70.8 |
| SPHINX-2k | - | - | - | - | - | 70.6 |
| LLaVA1.5 | - | - | - | - | - | 71.6 |
| GPT-3.5 (+Caption) | 70.14 | 62.43 | 68.18 | 78.59 | 52.32 | 67.18 |
| Cantor (GPT-3.5) | **73.45** | **83.38** | **88.64** | **84.31** | **66.55** | **77.54** |
| Gemini | 71.55 | 84.29 | **93.18** | 80.90 | 67.01 | 76.85 |
| Cantor (Gemini) | **79.49** | **86.39** | **93.18** | **86.98** | **71.26** | **82.40** |

generator and answer generator, Cantor reaches an accuracy of 84.96%, significantly surpassing all training-free methods, and even outperforming fine-tuned methods like UnifiedQA (CoT) [52] and MM-CoT [52]. This not only demonstrates the generality of Cantor but also shows that Cantor starts with perception-based information for making better decisions. Moreover, by invoking various expert

Table 3: Accuracy scores (%) on the *testmini* subset of MathVista, where bold entries indicate the best results, underlines indicate the second-best. Input: $Q$: question, $I$: image, $I_c$: image caption, $I_t$: OCR text detected in the image. Task types: FQA: figure question answering, GPS: geometry problem solving, MWP: math word problem, TQA: textbook question answering, VQA: visual question answering. Mathematical reasoning types: ALG: algebraic reasoning, ARI: arithmetic reasoning, GEO: geometry reasoning, LOG: logical reasoning, NUM: numeric commonsense, SCI: scientific reasoning, STA: statistical reasoning. ALL: overall accuracy. The performance results in the table come from [29].

| Model | Input | FQA | GPS | MWP | TQA | VQA | ALG | ARI | GEO | LOG | NUM | SCI | STA | ALL |
|---|---|---|---|---|---|---|---|---|---|---|---|---|---|---|
| *Heuristics baselines* | | | | | | | | | | | | | | |
| Random chance | - | 18.2 | 21.6 | 3.8 | 19.6 | 26.3 | 21.7 | 14.7 | 20.1 | 13.5 | 8.3 | 17.2 | 16.3 | 17.9 |
| Frequent guess | - | 22.7 | 34.1 | 20.4 | 31.0 | 24.6 | 33.1 | 18.7 | 31.4 | 24.3 | 19.4 | 32.0 | 20.9 | 26.3 |
| *Large Language Models (LLMs)* | | | | | | | | | | | | | | |
| Zero-shot GPT-3.5 | $Q$ only | 21.9 | 26.9 | 9.1 | 38.6 | 23.5 | 27.7 | 15.9 | 25.7 | 21.6 | 9.9 | 41.5 | 20.5 | 23.5 |
| Zero-shot GPT-4 | $Q$ only | 22.3 | 37.0 | 7.0 | 39.2 | 27.4 | 33.6 | 17.4 | 35.6 | 16.2 | 9.2 | 45.8 | 19.5 | 26.1 |
| Zero-shot Claude-2 | $Q$ only | 21.9 | 34.1 | 13.4 | 36.1 | 29.1 | 32.8 | 20.4 | 33.3 | 13.5 | 12.1 | 36.4 | 20.5 | 26.4 |
| *Augmented Large Language Models (Augmented-LLMs)* | | | | | | | | | | | | | | |
| 2-shot CoT GPT-3.5 | $Q, I_c, I_t$ | 27.5 | 29.3 | 36.0 | 49.4 | 29.1 | 31.0 | 32.9 | 31.0 | 16.2 | 17.4 | 50.8 | 37.2 | 33.2 |
| 2-shot CoT GPT-4 | $Q, I_c, I_t$ | 27.9 | 31.7 | 31.2 | 51.9 | 28.5 | 33.5 | 30.9 | 32.2 | 13.5 | 12.5 | **58.2** | 37.9 | 33.2 |
| 2-shot PoT GPT-3.5 | $Q, I_c, I_t$ | 24.5 | 26.4 | 23.7 | 33.5 | 27.9 | 27.8 | 26.1 | 28.0 | **18.9** | 13.2 | 33.6 | 29.9 | 26.8 |
| 2-shot PoT GPT-4 | $Q, I_c, I_t$ | 30.1 | **39.4** | 30.6 | 39.9 | 31.3 | **37.4** | 31.7 | **41.0** | **18.9** | 20.1 | 44.3 | 37.9 | 33.9 |
| GPT-3.5 | $Q, I_c$ | 26.0 | 31.7 | 35.5 | 48.1 | 30.2 | 32.4 | 32.3 | 33.0 | 16.2 | 17.4 | 54.9 | 36.2 | 33.2 |
| Cantor (GPT-3.5) | $Q, I_c$ | **45.7** | 31.8 | **40.9** | **55.1** | **44.1** | 34.5 | **42.2** | 33.9 | 13.5 | **36.1** | 55.0 | **55.5** | **43.1** |
| *Multimodal Large Language Models (MLLMs)* | | | | | | | | | | | | | | |
| IDEFICS-9B-Instruct | $Q, I$ | 21.6 | 21.1 | 6.5 | 25.9 | 24.0 | 22.1 | 15.0 | 19.8 | 18.9 | 9.9 | 24.6 | 18.1 | 19.8 |
| mPLUG-Owl-LLaMA-7B | $Q, I$ | 22.7 | 23.6 | 10.2 | 27.2 | 27.9 | 23.6 | 19.2 | 23.9 | 13.5 | 12.7 | 26.3 | 21.4 | 22.2 |
| miniGPT4-LLaMA-2-7B | $Q, I$ | 18.6 | 26.0 | 13.4 | 30.4 | 30.2 | 28.1 | 21.0 | 24.7 | 16.2 | 16.7 | 25.4 | 17.9 | 23.1 |
| LLaMA-Adapter-V2-7B | $Q, I$ | 21.2 | 25.5 | 11.3 | 32.3 | 31.8 | 26.3 | 20.4 | 24.3 | **24.3** | 13.9 | 29.5 | 18.3 | 23.9 |
| LLaVAR | $Q, I$ | 21.9 | 25.0 | 16.7 | 34.8 | 30.7 | 24.2 | 22.1 | 23.0 | 13.5 | 15.3 | 42.6 | 21.9 | 25.2 |
| InstructBLIP-Vicuna-7B | $Q, I$ | 23.1 | 20.7 | 18.3 | 32.3 | 35.2 | 21.8 | 27.1 | 20.7 | 18.9 | 20.4 | 33.0 | 23.1 | 25.3 |
| LLaVA-LLaMA-2-13B | $Q, I$ | 26.8 | 29.3 | 16.1 | 32.3 | 26.3 | 27.3 | 20.1 | 28.8 | **24.3** | 18.3 | 37.3 | 25.1 | 26.1 |
| Multimodal Bard | $Q, I$ | 26.0 | **47.1** | 29.6 | 48.7 | 26.8 | **46.5** | 28.6 | **47.8** | 13.5 | 14.9 | 47.5 | 33.0 | 34.8 |
| Gemini | $Q, I$ | 37.1 | 29.3 | 38.1 | **57.5** | 36.3 | 36.0 | 35.7 | 31.4 | **24.3** | 25.7 | **50.0** | 41.9 | 38.8 |
| Cantor (Gemini) | $Q, I$ | **50.2** | 39.4 | **39.8** | 49.4 | **43.8** | 42.0 | **41.5** | **41.4** | 10.8 | **30.8** | 46.7 | **59.5** | **44.7** |

modules, it can introduce richer contextual information to both LLMs and MLLMs, aiding in problem-solving.

Particularly noteworthy is that Cantor advances in the multimodal domain. As shown in Tab. 2, we further present the accuracy of various methods on ScienceQA for the IMG class, which includes image context. It can be seen that Cantor based on GPT-3.5 significantly surpasses the baseline in various problems, and even surpasses well-known MLLMs such as SPHINX [26] and LLaVA-1.5 [27]. This indicates that clear perceptual decisions can trigger the reasoning ability of language models toward dense image information. At the same time, the experiment on Gemini also shows that we further stimulate the visual reasoning ability of MLLM.

**MathVista.** MathVista [29] is a challenging dataset that integrating a variety of mathematical reasoning tasks with visual tasks. Tab. 3 compares different method performances. We also conduct experiments using GPT-3.5 and Gemini as baselines. From general visual question answering to professional math word problems,

Table 4: The impact of different levels of visual information on model's performance.

| Analysis | ScienceQA | MathVista |
|---|---|---|
| No Visual Information | 65.69 | 25.70 |
| + Rough Caption | 63.21 | 25.10 |
| + Detailed Caption | 74.37 | 33.20 |
| + Image | **78.85** | **38.00** |

Cantor has greatly surpassed the baseline in almost all types of problems. This indicates that correct decision and modular experts can stimulate their fine-grained, in-depth visual understanding and combinatorial reasoning abilities. It is worth noting that Cantor (GPT-3.5) even surpasses GPT-4 based on CoT and PoT.

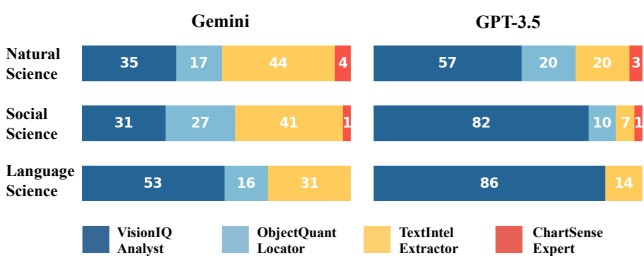

**Figure 3: Proportions of Cantor's invocation of expert modules across three types of questions on ScienceQA.**

## 4.5 Quantitative Analysis

**Analsis Visual Cues for Decision Generation.** We conduct a detailed analysis of the impact of visual information on Gemini's decision generation on ScienceQA and MathVista, with the prompt "think step by step". The results are shown in Tab. 4. When we do not input any form of visual information (including images and captions) in the experiment, only the text of the question is input. It can be seen that even without any visual information, MLLMs like Gemini still possess strong logical reasoning ability in pure language modal, demonstrating its superiority as a decision generator. Then we step by step explore the impact of incorporating visual information on Gemini. Firstly, we add rough captions, such as "A photo of a black and white cat." Gemini's performances unexpectedly decline on both datasets. This indicates that overly simplistic captions not only fail to promote MLLM, but can even mislead them into making incorrect decisions. Next, we enrich the description of captions to fully reproduce the image scene as much as possible. It can be seen that with the addition of detailed captions, Gemini's performance has significantly improved compared to those without visual information or rough captions. This indicates that visual information is indispensable for complex visual reasoning tasks. Finally, we replace captions with images, and it can be seen that Gemini's performance increased by 4.48% and 4.8% on both datasets, achieving the best performance at the same time. This is also in line with intuition, as the generation of captions is uncontrollable and may not necessarily contain key information for solving problems, but images themselves must have complete information. Therefore, in complex visual reasoning tasks, using images instead of captions to obtain visual information is a better solution for MLLM.

**Expert Module Use Planning.** The proportion of Cantor calling various expert modules on ScienceQA is shown in Fig. 3. We find that GPT-3.5 and Gemini exhibit different decision-generating behaviors. GPT-3.5 has a strong preference for using Object Quant Locator, with usage rates exceeding 80% in both Social Science and Language Science subjects, far exceeding other expert modules. We speculate that this is because GPT-3.5 is heavily influenced by in-context examples. On the other hand, Gemini is relatively balanced in expert module calls and does not exhibit any particular preferences. In addition, the usage ratio of both modules for ChartSense Expert is very low, especially for the Language Science subject where the number of calls is 0. This is because the proportion of questions related to table content is very small in ScienceQA, and there is even no question about table content in Language Science.

**Table 5: Performance increase with enabled modules and performance drop with disabled modules on ScienceQA, where "Enable Only" only just this module is on, others off. "Disable Only" means just this module is off, others on. In the last line, "Gemini/Cantor" denotes the original Gemini baseline and the fully implemented version of Cantor.**

| Module | Enable Only | Disable Only |
|---|---|---|
| TextIntel Extractor | 80.91(+4.06) | 80.86(-1.54) |
| ObjectQuant Locator | 80.27(+3.42) | 81.01(-1.39) |
| VisionIQ Analyst | 80.22(+3.37) | 81.51(-0.89) |
| ChartSense Expert | 79.13(+2.28) | 81.71(-0.69) |
| Gemini / Cantor | 76.85 | 82.40 |

This demonstrates the rationality of the decisions made by the two models. For different types of problems, the Language Science subject focuses more on the language meaning behind the image rather than being limited to the combination of target numbers or positions. Therefore, the two models call VisionIQ Analyst more frequently, reducing the use of ObjectQuant Locator.

**Ablation Study with Modules.** We use Gemini as the MLLM to investigate the impact of enabling and disabling expert modules on the performance of ScienceQA. The results are shown in Tab. 5. The results show that the use of each expert module results in a gain (maximum 4.06%, minimum 2.28%), indicating that all expert modules play a crucial role. The TextIntel Extractor is the most important among all modules, with the most significant gains and decreases in performance. At the same time, we can also find that enabling a module has a greater impact on model performance than disabling it. We believe that the effective high-level information obtained by an expert module(MLLM) is more generalized, compared with lower-level visual-information (such as coordinates, color, attributes, etc.). This higher-level information assists in the execution of other module tasks. In our method, even if a module is disabled, MLLM playing the role of other experts can to some extent compensate for the lack of that module, as they are not operating in isolation. We have also added some results in the supplementary material to support this view.

## 5 CONCLUSION

In this paper, we introduce an inspiring multimodal chain-of-thought framework named Cantor, designed to enhance the determining capabilities of MLLMs. By delving into the pivotal role of visual information in the decision-generating process, this paper highlights the importance of integrating visual cues at the decision stage, effectively mitigating the hallucination issues that may arise in LLMs. The novelty of the Cantor framework also lies in its ability to enable an MLLM to emulate the roles of domain-specific experts, acquiring high-level information, and thereby facilitating more rational and in-depth reasoning processes. Demonstrated on the challenging benchmarks of ScienceQA and MathVista involving complex visual reasoning tasks, Cantor has shown remarkable adaptability and efficacy, proving its strong potential in addressing real-world reasoning problems across various domains.

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
