# OpenReview forum: "Cantor:  Inspiring Multimodal Chain-of-Thought of MLLM"
_acmmm.org/ACMMM/2024/Conference — MM2024 Poster_

### Official Review · Reviewer_2Zrj · 2024-05-23

**Rating:** 3
**Confidence:** 4

**Summary:**

This paper introduces Cantor, a training-free CoT method to enhance the tool usage ability of MLLM. The core of the Cantor system is a Decision-Generation process with a suite of specialized expert modules.

**Strengths:**

1. This paper is generally well-structured and articulated.

2. The motivation is clear and well-founded. Moreover, the comparison of providing only-text information and visual contexts is convincing.

3. The training-free methods and comparable performance with some fine-tuning methods can possibly inspire some CoT studies.

**Limitations:**

1. The experimental section could benefit from additional comparisons with zero-shot or few-shot of some recent closed-source LMMs, e.g., GPT4v, Gemini1.5, Claude3, and the performance of integrating with Cantor.

2. Cantor's application may be limited since there are only four modules. If Cantor encounters problems that are beyond the scope of the four modules, how will it deal with the situation, and what outputs will it make? And if users want to add more function modules afterwards, e.g. code analyst, can Cantor extend to new modules conveniently?

3. A comparison with works using external tools would enhance the study, like Visual ChatGPT[1], HuggingGPT[2]. Besides that, a similar work MLLM-Tool[3] seems to explore the influence of visual content on the tool usage performance as well, what's the difference between Cantor and MLLM-Tool.

4. I do not think the rough caption is meaningful enough. From the user's point of view,  the proposed caption would only be very concise instead of not having any information. The more meaningful comparison may be between the annotation made by humans and the detailed caption.

5. The experiment ablated on the number of modules used can illustrate the effect of the difficulty of cases.

Overall, I recommend Borderline Reject, but I am also open to hearing how authors respond to those weaknesses mentioned above, and I'm available to raise my rating.

[1]HuggingGPT: Solving ai tasks with chatgpt and its friends in huggingface
[2]VisualChatGPT: Talking, drawing and editing with visual foundation models
[3]MLLM-Tool: A Multimodal Large Language Model For Tool Agent Learning

**Suitability:**

3

---

### Official Review · Reviewer_9cYS · 2024-05-24

**Rating:** 4
**Confidence:** 4

**Summary:**

In this paper, authors propose a novel multimodal CoT framework. It enables LLMs recognising and organising higher level of logical information when handling visual reasoning task. To better handle the visual information, CoT introduces the mechanism of tailored experts to tickle the weakness on solving problems requiring multiple steps. CoT provides an effective and feasible solution for complicated visual reasoning problems. In this paper, the experiment results demonstrate the claimed strength of CoT. The solid promotion of performance on challenging benchmarks provide a convincing demonstration of CoT.

**Strengths:**

1. This paper provides an intuitive and novel solution to enhance the ability of existing LLMs (MLLMs) on logical information recognition in various visual reasoning task. CoT proposes a comprehensive step-by-step decision generation process. The design innovates breaking the given problem into multiple component to avoid the limitation of LLM on complicated visual reasoning.



2. The proposed expert modules in execution step utilize multiple existing modules to resolve individual decomposition problems independently. It benefits the overall recognition performance on the higher level information retrieval.



3. The experiments are implemented on two challenging benchmarks with wide range of existing methods. The results provide a convincing demonstration on performance promotion brought by CoT. The ablation study also provides a comprehensive insight on proposing pipeline.

**Limitations:**

Below are some limitations and suggestions for the paper.

1. In Section 3, it seems that the subtasks and corresponding answers are the most important roles in the proposed method. However, the author didn't explain how to decide the number $n$ (In supply material, $n$ will change in different cases). Additionally, the author didn't report the impact of $n$ in the ablation study. Please explain (1) how to decide $n$ number?  (2) whether a bigger number $n$ will make performance better?

2. The paper may also consider directly reporting the performance of GPT-4V (or maybe GPT-4o) in Tables 1, 2 and 3 for reviewers to have a better comparison since the author already compared with LLaVA and LLaVA 1.5 (MLLM models).

3. The authors report that Cantor (GPT-3.5) even surpasses GPT-4 based on CoT and PoT, which is impressive. However, it would be more convincing to also report Cantor based on GPT-4 (Since GPT-4 is a stronger base model against GPT3.5) and compare it against GPT-4V.

**Suitability:**

3

---

### Official Review · Reviewer_VzN8 · 2024-05-25

**Rating:** 2
**Confidence:** 3

**Summary:**

The text build a framework Cantor, this is a novel multimodal CoT (Context of Thinking) framework designed to address various limitations in decision generation tasks. Cantor employs both Multimodal Language Models (MLLM) and Language Models (LLM) to process visual and textual contexts simultaneously for comprehensive understanding. During decision generation, Cantor utilizes MLLMs or LLMs as "experts" to handle specific tasks, with MLLMs being more adept at high-level logical problem-solving. Finally, the proposed method is effective on two challenging benchmarks, largely surpassing existing approaches.

**Strengths:**

1)	This work introduces an inspiring multimodal CoT framework to improve the determining capabilities of existing large multimodal models.
2)	The novelty Cantor strategy can inspire an MLLM to emulate the roles of domain-specific expert and obtain more in-depth reasoning process about the large model.
3)	The experiments are comprehensive, comparing the effectiveness of Cantor with previous methods, and ablation studies provide evidence to validate the efficacy of the proposed approach.

**Limitations:**

1)	In L157-158, the authors argue that "we conclude that visual information is crucial during the decision generation stage" is a new conclusion. However, similar insights can be found in many previous works, such as [1]. Moreover, I believe this conclusion is normal and lacks new contributions to our community.
2)	The proposed Cantor framework decouples a complex reasoning problem into multiple different sub-tasks with different modules, and this operation lacks innovation. Similar ideas can be found in many previous works, such as [2-3].
3)	From Table 1 and Table 3, compared to the proposed complex execution process of Cantor, the performance improvement is marginal. On multiple tasks, Cantor even yields poorer results than the basic Gemini.

[1] Multimodal chain-of-thought reasoning in language models
[2] Chameleon: Plug-and-play compositional reasoning with large language models
[3] Learning to Reason: End-To-End Module Networks for Visual Question Answering

**Suitability:**

3

---

### Meta-Review · Area_Chair_4zFA · 2024-07-02

**Recommendation:** Accept (Poster)
**Confidence:** 5

**Metareview:**

Reviewers have relatively borderline scores. The main concerns are the similar contributions as prior works, some implementation details, the extension ability of the method, and the results on latest GPT-4o. The author rebuttal addressed some of the concerns, while reviewer VzN8 still has doubts about the contribution. After checking the text, the AC thinks this work presents a novel planning method for complex multimodal question answering problem, which has enough contributions.